# Lattice reconstruction induced multiple ultra-flat bands in twisted bilayer WSe$_2$

En Li[1,2], Jin-Xin Hu[1,2], Xuemeng Feng[1,2], Zishu Zhou[1], Liheng An[1], Kam Tuen Law [1✉], Ning Wang [1✉] & Nian Lin [1✉]

Moiré superlattices in van der Waals heterostructures provide a tunable platform to study emergent properties that are absent in the natural crystal form. Twisted bilayer transition metal dichalcogenides (TB-TMDs) can host moiré flat bands over a wide range of twist angles. For twist angle close to 60°, it was predicted that TB-TMDs undergo a lattice reconstruction which causes the formation of ultra-flat bands. Here, by using scanning tunneling microscopy and spectroscopy, we show the emergence of multiple ultra-flat bands in twisted bilayer WSe$_2$ when the twist angle is within 3° of 60°. The ultra-flat bands are manifested as narrow tunneling conductance peaks with estimated bandwidth less than 10 meV, which is only a fraction of the estimated on-site Coulomb repulsion energy. The number of these ultra-flat bands and spatial distribution of the wavefunctions match well with the theoretical predictions, strongly evidencing that the observed ultra-flat bands are induced by lattice reconstruction. Our work provides a foundation for further study of the exotic correlated phases in TB-TMDs.

[1] Department of Physics, The Hong Kong University of Science and Technology, Hong Kong SAR, China. [2] These authors contributed equally: En Li, Jin-Xin Hu, Xuemeng Feng. ✉email: phlaw@ust.hk; phwang@ust.hk; phnlin@ust.hk

Single-layer two-dimensional (2D) materials have different physical and chemical properties from bulk materials[1–3]. When vertically stacking two monolayers of 2D materials with a twist angle (or 2D materials with different lattice constants), moiré superstructures will form and such periodic potential can dramatically modulate the electronic and optical properties[4–7]. In recent years, an important breakthrough was the experimental observation of correlated insulator and super-conductivity in twisted bilayer graphene (TBG)[8,9]. Near magic twist angles, flat bands emerge near the Fermi level in TBG and promote electron-electron interactions, resulting in strongly correlated phases, such as superconductivity and correlated insulators[8–11].

Inspired by the discoveries in TBG, flat bands have been predicted in other twisted bilayers with moiré superstructures[12–14], especially in twisted bilayer transition metal dichalcogenides (TB-TMDs)[14,15]. Unlike the flat bands in TBG which only emerge at certain twist angles, flat bands with bandwidth about tens of meV appear in a wide range of twist angles in twisted bilayer TMDs[14,16]. Since TMDs have large effective masses and relatively strong electron-electron interactions[17,18], varied flat-band behaviors with twist angle make TB-TMDs a highly tunable platform to study correlated physics[15,16,19]. For example, novel topological insulating states have been predicted to emerge in TMD heterobilayer[20] and twisted homobilayers[21]. Evidence of low-energy flat bands and correlating physics, including correlated insulators and generalized Wigner crystal, have been observed in twisted bilayer TMDs by both transport and optical studies[22–26]. Despite these exciting progresses, there is an urgent need to understand the properties and underlying physical origin of the flat bands.

In this work, we observed multiple ultra-flat bands near valence band (VB) edges in twisted bilayer $WSe_2$ (TB-$WSe_2$) with various twist angles around 60°, by low-temperature scanning tunneling microscopy and spectroscopy (STM/STS) measurements. These ultra-flat bands, with bandwidths estimated to be a few meV only, appear when the twist angle is within 3° of 60°. The number of the flat bands increases and the bandwidth decreases as the twist angle approaches 60°. Combining with theoretical modeling, we conclude that in addition to the well-known basic moiré bands caused by interlayer hybridization, the ultra-flat bands mainly arise from atomic reconstruction of the moiré superlattices when the twist angle is within 3° of 60° due to the enhanced interlayer interaction. The ultra-flatness of the bands in the order of meV, makes this system highly susceptible to electron–electron interactions induced effects. Our work lays the foundation for the understanding of the emergence and tunability of ultra-flat bands in bilayer $WSe_2$, which is essential for the understanding of correlated phases in these materials.

## Results and discussion

**Electronic structure without lattice reconstruction in the sample with 54.1° twist angle.** Different from TBG, twisted bilayer TMDs form two distinct moiré structures for the twist angle ($\theta$) close to 0° and 60°[14,16], owing to sublattice symmetry breaking in the TMDs. The distinct high-symmetry stackings make the electronic structure for twist angle around 0° and 60° significantly different[14,16], for example, the different spatial distribution of the flat-band wavefunctions in 3° and 57.5° TB-$WSe_2$[27]. In this work, we focus on the samples with the twist angle around 60° due to the following: it has been shown by theoretical analysis[28,29] and transmission electron microscopy (TEM) observations[30,31] that, at a small misalignment angle, the moiré superlattices undergo a strongly structural reconstruction rather than rigidly rotated crystalline lattices. The reconstruction

necessarily modifies the electronic and excitonic properties of the twisted bilayers[32–36]. Although this twist angle-dependent reconstruction happens in either near 0° or 60° TB-TMDs, calculations demonstrate that flat bands in near 0° TB-TMDs mainly originate from inhomogeneous hybridization, regardless of the reconstruction. In contrast, for the twist angle close to 60°, local strains in reconstructed moiré superlattices play an important role in engineering the moiré potential, resulting in multiple energy-separated ultra-flat bands[16]. However, experimental evidence for these ultra-flat bands is still missing. Moreover, TB-TMDs with twist angle around 60° is also a platform to study Hubbard model physics, which have been realized in TMD heterostructure[25]. Herein, series of TB-$WSe_2$ with various twist angles around 60° have been fabricated and investigated. The samples cover the transition from un-reconstructed lattice to reconstructed lattice, which allows us to distinguish the physical origin of the flat bands in TB-$WSe_2$ with and without lattice reconstruction.

Figure 1a shows an optical microscope image of a 54° TB-$WSe_2$ sample, in which lattice reconstruction does not occur. Two pieces of monolayer $WSe_2$ sit on a highly oriented pyrolytic graphite (HOPG) substrate with a rotation angle of 54°. The stacked region is marked by a red-dashed box, confirming that the stacking region only consists of bilayer $WSe_2$. In the bilayer region, a uniform moiré pattern is resolved by STM, as shown in Supplementary Fig. 1a. An atomic-resolution STM image shows the topmost Se atoms of $WSe_2$ with a moiré corrugation (Fig. 1b). Based on the measured moiré period ($\ell$:~3.2 nm) and atomic lattice (~0.33 nm), the twist angle can be identified to be 54.1°, in good agreement with the alignment angle determined in the sample fabrication. The appearance of the moiré corrugation in STM images originates from various high-symmetry stackings. We label the spatial locations as O, A, B (2H stacking), and Br (bridge between A and B). Their corresponding stackings are illustrated at the bottom of Fig. 1b, according to the structural modeling[14] and previous STM/STS analysis[27]. We carried out the STS measurements to investigate the electronic structure in this moiré superlattice. Supplementary Figure 1b is the large-range logarithm of d$I$/d$V$ spectra acquired at the center of O, A, B, and Br. These spectra consistently show a nearly intrinsic semiconductor bandgap of 2.04 eV. A careful inspection of the site-specific spectra and the spatially resolved conductance map in Supplementary Fig. 1c reveals that the VB edge shows moiré dependent features.

We focus on the VB edge. As shown in Fig. 1c, B, A, and Br sites feature different d$I$/d$V$ spectra. Site B shows a pronounced peak at −1.14 V (v1) with a full width at half maximum (FWHM) of ~37 meV. Site A shows a peak at −1.17 V (v2). As the bridge connecting A and B, Br sites show the residual of the first two peaks and raise the third peak at −1.21 V (v3). To elucidate the spatial distribution of the electronic states with respect to the moiré pattern, we acquired a series of STS spectra across the four high-symmetry sites along the green line in Fig. 1b, presented as the line-mode conductance map displayed in Fig. 1d. The v1 (v2) state is mostly distributed in the B (A) region, while the v3 state is located at the Br site. To further resolve the spatial distribution of these electronic states, we performed 2D spectroscopic mapping near the VB edge. Figure 1e shows the 2D maps at the selected energies (v1–v3) of the same surface area, showing that the v1 and v2 states are distributed in the B and A regions, respectively. The v3 mainly lies at the Br site, displaying a Kagome-like lattice structure as indicated in Fig. 1e(v3).

To understand the origin of the spectral peaks near the VB edge depicted in Fig. 1c, we calculated the electronic structure using a continuum model for the Γ valley moiré structure of TB-TMDs (Methods section). Figure 1f is the calculated local density

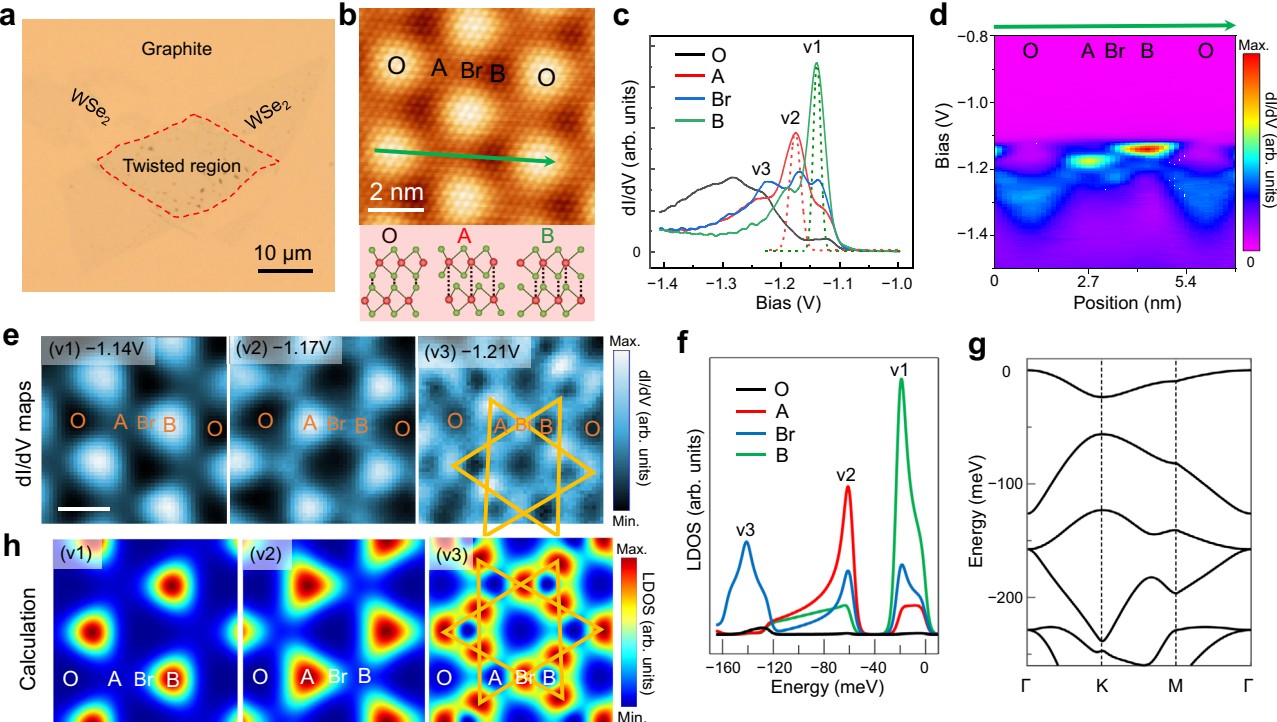

**Fig. 1 STM/STS measurement on 54.1° TB-WSe₂. a** Optical image of 54.1° TB-WSe₂ sample. The red-dashed box highlights the twisted region. **b** Top: an atomic-resolution STM topographic image (−1.5 V, 600 pA) of 54.1° TB-WSe₂. Bottom: a schematic side view of stacking configurations between two WSe₂ layers: O, A, B. Green: Se; Red: W. **c** $dI/dV$ spectrums ($V = -1.5$ V, $I = 500$ pA, $V_{mod} = 10$ mV) near VB edge acquired at the center of four moiré sites. The dashed green (red) curve is the estimated intrinsic v1 (v2) peak from Gaussian fitted peak in region B (A), respectively. **d** Conductance map taken along the green line in panel **b**, $V = -1.5$ V, $I = 1$ nA, $V_{mod} = 10$ mV. **e** $dI/dV$ maps at −1.14 V (v1), −1.17 V (v2) and −1.21 V (v3). Scale bar: 2 nm. **f** Calculated LDOS at four moiré sites, showing the site-dependent peak intensity in moiré lattice. **g** Calculated electronic band structure plotted in the folded moiré Brillouin zone. Note that the valence band edge is set at $E = 0$. **h** Calculated LDOS maps at the energy of v1, v2, and v3, respectively, agree well with experimental results shown in **e**.

of states (LDOS) at four moiré sites, which match very well with the measured site-specific $dI/dV$ spectrums (Fig. 1c, d). The calculated energy separation between v1 and v2 is ~40 meV, agreeing well with experimental result (~36.5 mV). Figure 1g shows the calculated band structure which reveals a series of moiré bands with bandwidths in the order of tens of meV (v1: ~30 meV, and v2: ~60 meV). Since the calculated v1 is an isolated flat band, we further estimate the intrinsic v1/v2 peak from measured $dI/dV$ spectrums. As described in Supplementary Note 2, the deconvoluted intrinsic v1/v2 peak is shown in dashed curves in Fig. 1c, indicating a nearly isolated v1 peak. Furthermore, the spatial distributions of the calculated density of states at the energy of the spectral peaks are displayed as 2D LDOS maps shown in Fig. 1h, agreeing very well with measured 2D $dI/dV$ maps (Fig. 1e). For 54.1° TB-WSe₂, we estimate the on-site Coulomb repulsive interaction $U$ by using $U = e^2/(4\pi\epsilon d) \approx 180$meV. Here d ≈ 2nm is the spatial extent of the localized wavefunction, determined from $dI/dV$ maps in Fig. 1e, and $\epsilon = 4$ is the in-plane dielectric constant. Such a reasonably large ratio for $U$/bandwidth suggests the possibility of correlated insulator phase at half-filling or fractional-filling of the first moiré band[25,37].

**Lattice reconstruction at twist angle within 3° of 60°.** To explore the evolution of moiré-mediated bands against the twist angle, we fabricated and measured the TB-WSe₂ samples with twist angles of 57°, 57.4°, 57.8°, and 58.4°. For each moiré pattern, we systematically carried out spatial-dependent STS measurements. Figure 2b, e, h, k display the $dI/dV$ spectrums near the VB edge

acquired at the center of three moiré sites, revealing the isolated sharp band-edge peaks (in green) at B sites near the VB edge. Figure 2c, f, i, l show the line-mode conductance maps acquired point-by-point along the green line in the corresponding STM image in Fig. 2, displaying the evolution of the electronic states near the VB edge at gradually increased twist angles. The 57° TB-WSe₂ ($\ell$: ~6.3 nm) features two states (v1 and v2) near the VB edge that are separated by ~66 meV in energy, as marked by two white arrows in Fig. 2c. Both states are localized in region B but their spatial distributions are very different: v1 is distributed at the center region while v2 is distributed at the two sides having a central "node". As the twist angle further increased to 57.4° ($\ell$: ~7.2 nm) and 57.8° ($\ell$: ~8.6 nm), we observed three (v1, v2, and v3) and four (v1, v2, v3, and v4) energy-separated states that were localized in the region B, respectively. The distributions of v1 and v2 are similar to those of the 57.0° sample. The v3 state has two nodes and the v4 state has three nodes. As will be shown below, these nodes are related to the quantum well states caused by lattice reconstruction. For the 58.4° TB-WSe₂ sample ($\ell$: ~11.8 nm), a sequence of states emerges in region B as shown in Fig. 2k, l. Apart from the increased states numbers, the energy separation of observed states decreases as the twist angle approaches 60°. Note that the slight concave-up streaks of these states distribution are likely due to tip-induced band bending (TIBB) effects, similar behavior has been observed in aligned MoS₂/WSe₂ heterobilayer[38].

The continuum model with the $V_M(r)$ moiré potential (Methods) can well describe the states in 54.1° TB-WSe₂, however, we find it fails to interpret the quantum-well-like states observed in the 57°-58.4° samples (Supplementary Fig. 5). This

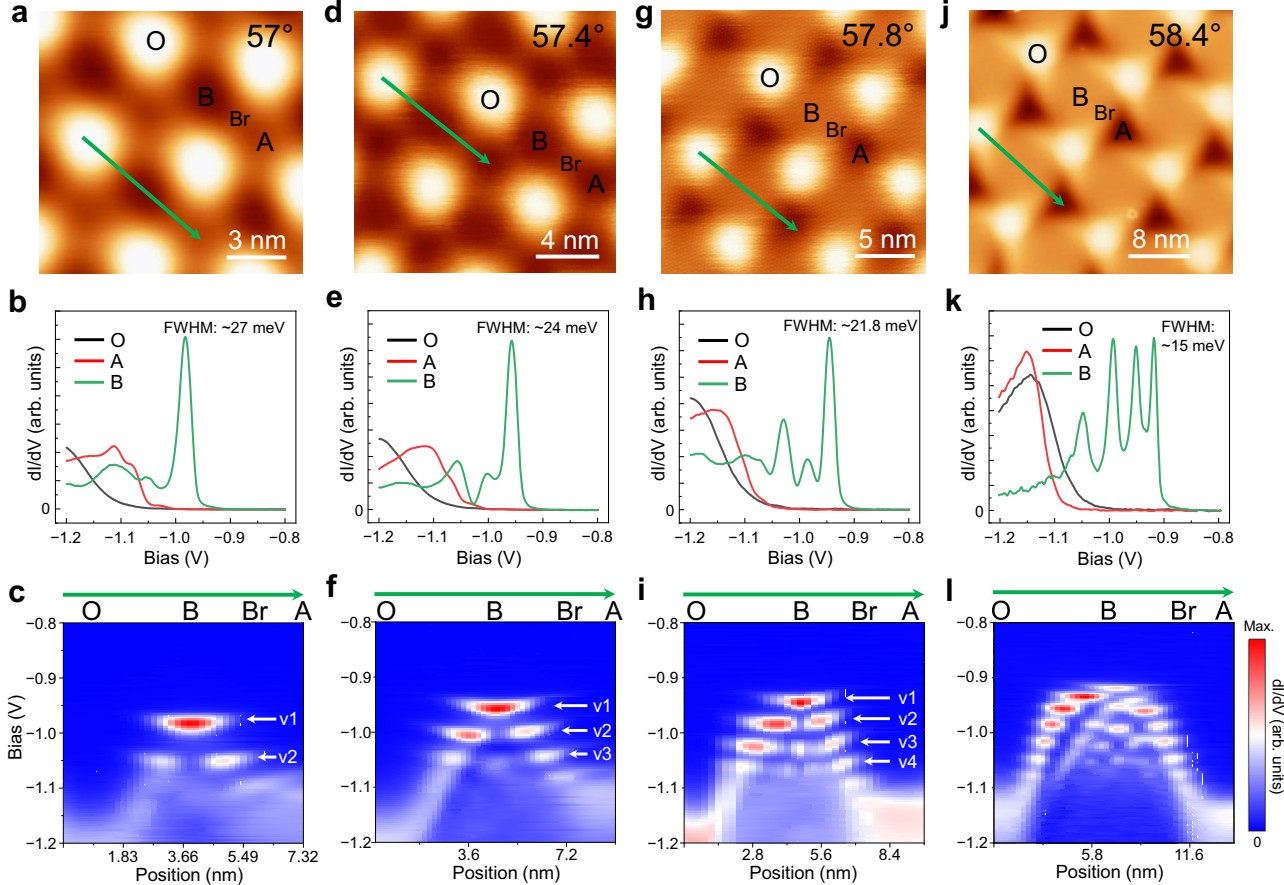

**Fig. 2 Spatially dependent spectroscopy of 57°, 57.4°, 57.8°, and 58.4° TB-WSe₂. a** An STM topographic image (−1.4 V, 1.0 nA) of 57° TB-WSe₂. **b** d$I$/d$V$ spectrums near VB edge acquired at the center of three moiré sites, $V = -1.4$ V, $I = 1$ nA, $V_{mod} = 10$ mV. The FWHM is extracted from a Gaussian fit applied to the data. **c** Conductance map taken along the green line in panel **a**, revealing two B-confined states (marked by white arrows). Similarly, **d-f** for 57.4°, $V = -1.4$ V, $I = 1$ nA, $V_{mod} = 10$ mV; **g-i** for 57.8°, $V = -1.4$ V, $I = 1$ nA, $V_{mod} = 10$ mV; **j-l** for 58.4°, $V = -1.3$ V, $I = 1$ nA, $V_{mod} = 5$ mV.

indicates that the multiple ultra-flat bands in the 57°–58.4° samples have distinct origins from the 54.1° sample. As revealed by early works[28,30], for $\theta > 57°$, TB-TMDs feature lattice reconstruction and domain formation. Calculations demonstrate that local strains caused by the reconstruction result in the formation of the triangular potential wells in moiré structure, which will give a series of localized quantum-well states[16]. Therefore, we adopt a reconstructed confining potential ($V_C$) to describe the atomic reconstruction and rewrite the continuum model, as described in Methods section and Supplementary Note 4. Figure 3 shows the calculated band structure (left) and the LDOS distribution across the O-B-A-O sites (right), for 57° (**a**), 57.4° (**d**), 57.8° (**g**), and 58.4° (**j**) TB-WSe₂. Multiple energy-separated flat bands emerge at the VB edge and are nearly dispersionless with bandwidth <1 meV. These ultra-flat bands are confined in region B, showing the characteristics of the quantum-well-like states. The numbers and energy separation of ultra-flat bands evolve with the twist angles. The relative energy levels of the experimentally resolved quantum-well-like states are plotted in orange lines. One can see that the energy levels, as well as the spatial distributions of the quantum-well-like states, agree well with the experimental data shown in Fig. 2c, f, i, l.

**Ultra-flat bands induced by lattice reconstruction.** Here we discuss the bandwidth of the flat bands. The calculated flat bands for twist angles greater than 57° are nearly dispersionless, while 54.1° TB-WSe₂ hosts a flat band with a bandwidth of ~30 meV (v1). In our STS measurement, the energy resolution is given by

$\triangle E = \sqrt{(3.5kT)^2 + (2.5V_{mod})^2}$ [38]. Since we use a modulation voltage of $V_{mod} = 10$ mV for 54.1°–57.8° and 5 mV for 58.4° to distinguish peaks, the corresponding energy resolution at 5.3 K is 25 meV and 12.6 meV, respectively. For the 54.1° sample, a slightly larger peak width (FWHM: ~37 meV) could be attributed to the small dispersion of the first band as shown in Fig. 1g. In contrast, for 57°–58.4° TB-WSe₂, many of B-confined peaks are seen to have width very close to the value of STS resolution. As shown in Fig. 2b, e, h, k, FWHM of the first band-edge peak in center of B region is ~27-21.8 meV for 57°–57.8° samples ($V_{mod} = 10$ mV) and ~15 meV for 58.4° ($V_{mod} = 5$ mV). Hence, the observed widths of these peaks are mainly produced by the modulation. By fitting the obtained tunneling spectra, as described in Supplementary Note 2, we estimate the intrinsic bandwidth of the ultra-flat bands in the 57°, 57.4°, 57.8°, and 58.4° twisted sample to be 10.2, 8.8, 7.5, and 5.8 meV, respectively. Such narrow bandwidth of these ultra-flat bands, which is only a fraction of the estimated on-site Coulomb repulsion energy, makes this system highly susceptible to electron–electron interaction induced effects.

Furthermore, we perform d$I$/d$V$ mapping to study the spatial distribution of the observed sharp peaks in Fig. 2. As shown in Fig. 3c, f, i, l, d$I$/d$V$ maps of the states are sequentially consistent for 57°–58.4° TB-WSe₂. We note that the wavefunctions of the flat bands at different energy have very different spatial distributions, associated with the observed nodes in line-mode conductance maps in Fig. 2. We have calculated the wavefunction

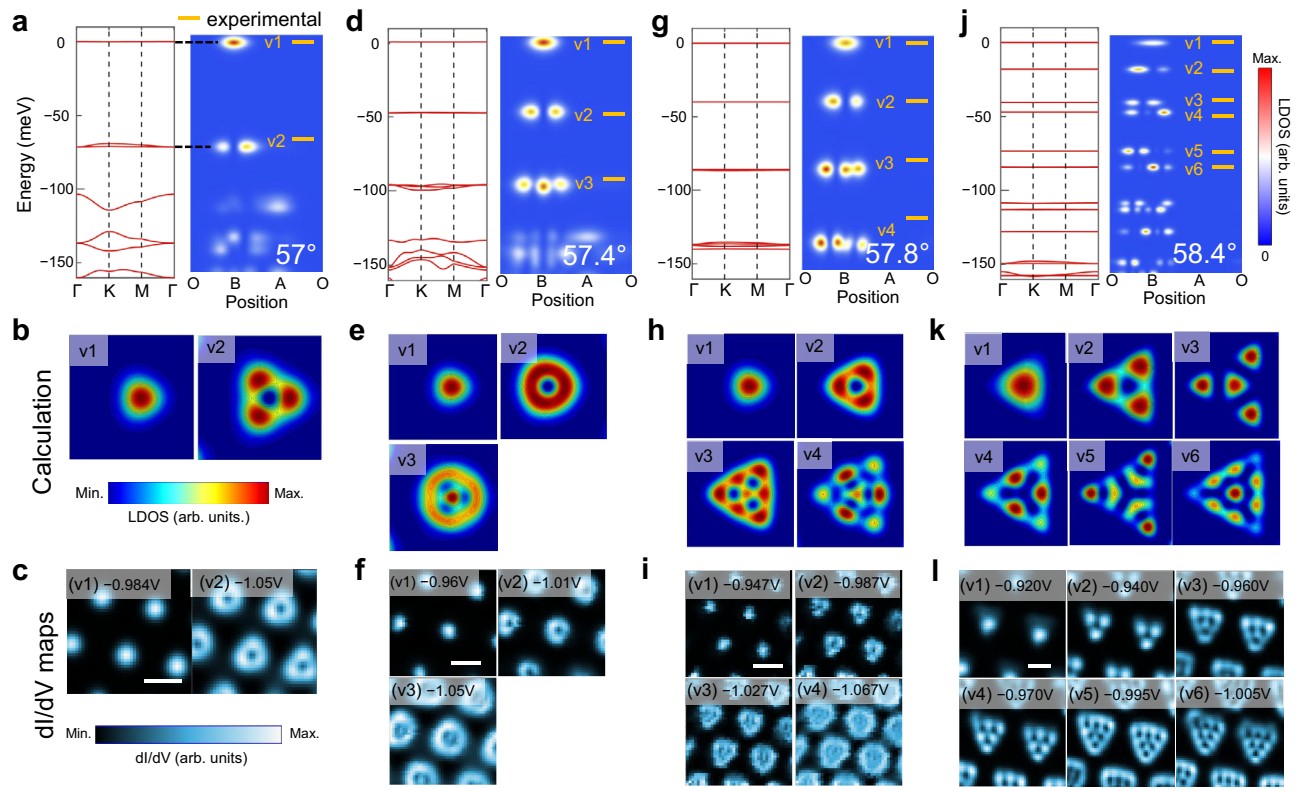

**Fig. 3 Evolution of calculated flat bands with twist angle. a** Calculated band structure (left) and the LDOS across the O-B-A-O sites (right) for 57° TB-WSe₂, showing the ultra-flat bands and corresponding spatial distributions. The orange lines mark the relative energy levels of the experimental states. Note that the valence band edge is set at $E = 0$. **b** Calculated wavefunction patterns of the flat bands labeled in **a**. **c** Measured dI/dV maps for 57° TB-WSe₂ at the energies of B-confined states in Fig. 2c. Scale bar: 4 nm. Similarly, **d–f** for 57.4°, scale bar in **f**: 4 nm; **g–i** for 57.8°, scale bar in **i**: 6 nm; and **j–l** for 58.4°, scale bar in **l**: 5 nm. **k** and **l** display the calculated and experimental LDOS maps of first six states in 58.4° TB-WSe₂, respectively.

patterns of the ultra-flat bands with lattice reconstruction and the results are depicted in Fig. 3b, e, h, k, showing that v1 appears as a dot at the center of region B, v2 appears as a three-dot triangle, v3 and v4 comprise more dots following the triangular symmetry. The calculated spatial distribution of the wavefunctions matches well with the dI/dV maps. This provides direct and convincing experimental evidence that the ultra-flat bands are originated from the lattice reconstruction.

**Lattice reconstruction-induced evolution of the morphology, moiré potential, and flat bands**. The exceptionally high degree of agreement between the STS measurements and the modeling suggests that the flat bands in the TB-WSe₂ can be attributed to the emergence of a moiré potential associated with lattice reconstruction. For the 54.1° sample, its potential term $V_M(\mathbf{r})$ is from inhomogeneity of the interlayer hybridization due to out-of-plane deformation in the moiré superlattice. As the twist angle further approaches 60°, lattice reconstruction happens when energy gain from the formation of favorable stacking overcomes elastic energy cost of strain produced by the local in-plane displacements[29]. Different from TBG and TB-TMDs around 0°, there is only one low-energy stacking (2H) in TB-TMDs with $\theta$ close to 60°[28,39]. In real space, the 2H stacking domains (B regions) expand overwhelmingly as twist angle approaches 60° and transform from an equilateral triangle to a Reuleaux triangle[16,28], as illustrated in Fig. 4a. In our STM images shown in Fig. 4b, the reduced size of O regions and increased B regions corroborates this phenomenon. This structural transformation of the moiré superlattice strongly influences the electronic structure, specifically, forming the multiple ultra-flat bands. Local strain caused by in-plane relaxations, results in the formation of

modulating confining potential[16]. As displayed in Fig. 4c, d, without this in-plane structural reconstruction (54.1°), the moiré potential profile modulates gradually and reaches maximum at the B site, which produces v1 state. The local maximum at the A site produces the v2 state. In sharp contrast, with the lattice reconstruction (58.4°), the potential profile features periodic quantum wells that give a series of confining states (v1, v2, v3, … etc.) in the B regions depending on the size and potential depth of the well. In contrast to the observed flat bands in the aligned TMD heterobilayers[33,38], which are localized at K or Γ points and mixed with other bands, the ultra-flat bands in this system are well separated in energy and show tunable bandwidth and energy level with twist angles, providing a platform to study the Hubbard model on a mesoscale triangular lattice.

The evolution of the moiré potential $V_M$ and confining potential $V_C$ against the twist angles used in our model calculations is shown in Supplementary Fig. 6. The gradual variation of the two potentials indicates a transition from the interlayer hybridization dominating to reconstruction-induced confining potential dominating. Moreover, we find that this transition accompanies the enhanced interlayer interaction in the bilayer WSe₂. As shown in Supplementary Fig. 2, the states away from VB/CB edges vary with the moiré structure. The peaks at ~ −1.8 V and 1.3–2.0 V show an energy downward shift at A sites. This downward shift becomes more pronounced as the twist angle changes from 54.1° to 58.4°, evidencing stronger interlayer interaction as the twist angle approaches 60°.

In summary, this study demonstrates two distinct mechanisms for the emergence of the flat bands in TB-WSe₂ with twist angles around 60°: At twist angles far from 60° (for instance 54.1°), the interlayer hybridization leads to two spatially separated flat bands

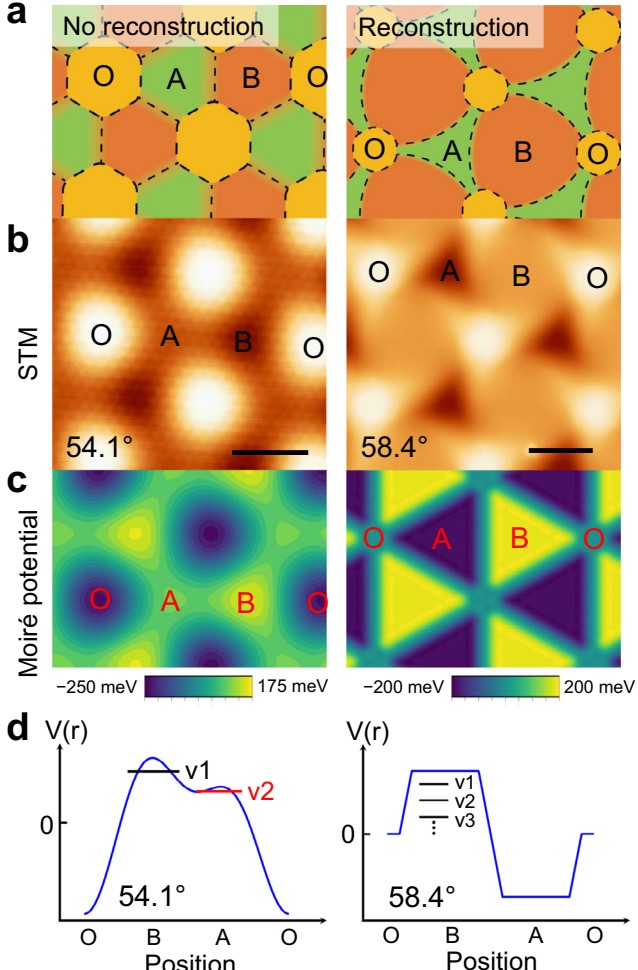

**Fig. 4 Moiré potential evolution due to lattice reconstruction. a** Illustration of the moiré pattern with and without twist angle-dependent reconstruction (right and left, respectively). Dashed lines mark the boundaries of O/A/B regions. **b** STM image of 54.1° (left, $V = -1.4$ V, $I = 1$ nA) and 58.4° (right, $V = -1.25$ V, $I = 1$ nA) TB-WSe$_2$, showing the topographic difference due to lattice reconstruction. The scale bar is 2 nm (54.1°) and 6 nm (58.4°), respectively. **c** Moiré potential for 54.1° (left) and 58.4° (right) TB-WSe$_2$. **d** Illustration of moiré potential profile for 54.1° (left) and 58.4° (right) TB-WSe$_2$ and the corresponding confined states.

with bandwidths in the order of tens of meV. As twist angle approaches 60° ($\theta > 57°$), lattice reconstruction accompanied with strong interlayer interaction creates the triangular quantum wells that host multiple energy-separated ultra-flat bands with bandwidths of only a few meV. The observed ultra-flat bands with tunable numbers and energy separation provide practical guidance for further realization of correlated states when the Fermi energy is properly tuned to the ultra-flat bands. We expect that the large on-site Coulomb repulsion energy to bandwidth ratio will result in interesting correlated phases in these reconstructed moiré systems.

## Methods

**Sample fabrication.** Our 54°–58.4° TB-WSe$_2$ samples were fabricated by the "tear and stack" method[40], based on the PDMS stack and the precisely rotational control stage (Mechanical accuracy is ±0.5°). The WSe$_2$ bulks are bought from HQ Graphene. The monolayer WSe$_2$ flakes with large size are exfoliated onto the SiO$_2$/Si wafer, which always have natural cracks, convenient for the same flake "tear and stack" process. Use the polydimethylsiloxane (PDMS) stack covered by the poly(bisphenol A carbonate) (PC) thin film to tear half (or part) of the monolayer flake, and stack onto the remnant part monolayer flake with a controlled interlayer

twist angle near 60°. Then, the TB-WSe$_2$ structures are transferred onto the freshly cleaved graphite substrate with alignment marks, serving as the guides for STM tip positioning.

**STM measurements.** STM/STS experiments were carried out in a commercial ultra-high vacuum low-temperature STM system (CreaTec) with a base pressure of $1.0 \times 10^{-10}$ mBar. All the STM/STS measurements were performed at 5.3 K using a chemically etched tungsten tip. Before STM measurement, the samples were annealed at ~ 250 °C for over 3 h to remove possible adsorbates. STM images were acquired in the constant-current mode and the bias voltages refer to the sample with respect to the STM tip. The d$I$/d$V$ spectra were collected by using the standard lock-in technique with a voltage modulation of 5–10 mV and frequency of 787.3 Hz. For each sample, STS measurements were performed at several areas of twisted region, to ensure the consistency and accuracy of the findings.

**Theoretical modeling of moiré band structure.** We construct the continuum model for 60°-twisted WSe$_2$ bilayers. For 60°-twisted bilayer WSe$_2$, the valence band maxima is always at Γ point not K point[14]. Neglecting spin-orbit coupling, which vanishes at the Γ-point by Kramer's theorem, we obtain the following simple single-band **k·p** Hamiltonian:

$$H = \frac{\hbar^2 k^2}{2m^*} + V_M(r)$$

$$V_M(r) = 2V_0 \sum_i \cos(\boldsymbol{g_i} \cdot \boldsymbol{r} + \phi)$$

$m^*$ is the effective mass at Γ pocket and $V_0, \phi$ are parameters which describe the moiré potential term. And $\boldsymbol{g_i}$ are moiré reciprocal vectors. Such potential is produced by the out-of-plane deformation induced inhomogeneous interlayer hybridization. We choose a typical set of parameters $(m^*, V_0, \phi) = (1.2m_e, 40\text{meV}, 170°)$ for $\theta = 54.1°$ case, which reproduces the experimental LDOS structure both qualitatively and quantitatively.

As the twist angle further approaches 60°, in-plane structural reconstruction results in an additional confining potential with an equilateral triangular shape[16]. We use a confining potential to describe this reconstruction effect:

$$V_C(\boldsymbol{r}) = -2V_1 \sum_{n=1}^{\infty} \sum_i \frac{1}{n^2} \frac{\sin(na\omega)}{a\omega} \sin(n\boldsymbol{g_i} \cdot \boldsymbol{r})$$

We use $\omega = \frac{2\pi}{L}, L = \sqrt{3}a_m, a = 0.2L$ (see Supplementary Fig. 6f) to produce a triangular well-like potential with continuous transited boundaries, based on previous DFT calculations[16]. In a reconstructed moiré structure, the combined effect of reconstruction-induced $V_C(\boldsymbol{r})$ and inhomogeneous hybridization-induced $V_M(\boldsymbol{r})$ results in the flat-band formation. Therefore, in our calculations for 57°–58.4° cases, we have adjusted the parameters $(V_0, V_1)$ in $((V_M(\boldsymbol{r}) + V_C(\boldsymbol{r}))$ to fit the observed flat-band structures, as shown in Supplementary Fig. 6. Note that the selection of the parameters is not unique but to fit with our experiment for comparison.

The formula of local density of states:

$$\psi_{n\boldsymbol{k}}(\boldsymbol{r}) = \sum_{\boldsymbol{G}} u_{n\boldsymbol{k}}(\boldsymbol{G}) e^{i(\boldsymbol{k}+\boldsymbol{G}) \cdot \boldsymbol{r}}$$

$$D(\boldsymbol{r}, E) = \sum_{n,\boldsymbol{k}} |\psi_{n\boldsymbol{k}}(\boldsymbol{r})|^2 \delta(E - E_{n\boldsymbol{k}})$$

## Data availability
The data that support the plots within this article is available from the corresponding authors upon reasonable request.

## Code availability
The computer codes that support the findings of this study are available from the corresponding author upon reasonable request.

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

## Acknowledgements
This work is partially supported by the National Key R&D Program of China (2020YFA0309600), the equipment fund of Hong Kong UGC (C6012-17E), and the Research Grants Council of Hong Kong (Project No. RFS2021-6S03, C6025-19G, AoE/P-701/20, 16310520, 16310219, 16309718 and 16303720.). K.-T.L. acknowledge the support of the Ministry of Science and Technology (MOST20SC04) and the Croucher Foundation.

## Author contributions
N.L., N.W., and K.-T.L. supervised the work. E.L. performed the STM/STS experiments. X.F., Z.Z., and L.A. fabricated the samples. J.-X.H. computed the electronic structure under the supervision of K.-T.L.; E.L. and J.-X.H. analyzed the experimental data together with computational results. E.L., J.-X.H., N.L., N.W. and K.-T.L. wrote the manuscript with input from all other authors.

## Competing interests
The authors declare no competing interests.
