## [Peer Review File · Nature Communications]

REVIEWER COMMENTS

Reviewer #1 (Remarks to the Author):

The manuscript contains interesting information on a timely topic: the existence and characterization of flat bands in twisted transition metal dichalcogenide homo bilayers. The data quality is high, and the manuscript is clearly written. I am in favor of publishing it on Nature Communication.

With that said, I do not consider this study as a substantial breakthrough in the field, for the following reasons: 1) the sample is fabricated on a conductive graphene substrate, which is well known to quench the correlated phenomena. The quantum confinement like features in the gamma flat band has been previously reported in the CVD growth samples (eg, Pan, Y. et al. Quantum-confined electronic states arising from the moiré pattern of MoS₂-WSe₂ heterobilayers. *Nano Lett.* 18, 1849–1855 2018; Zhang, C. et al. Interlayer couplings, moiré patterns, and 2D electronic superlattices in MoS₂/WSe₂ hetero-bilayers. *Sci. Adv.* 3, e1601459 2017). @) Compare with these studies, the main highlight of the manuscript is the ability to control the twisted angle using mechanical transfer technique, which has also been reported previously (Zhang, Z. et al. Flat bands in twisted bilayer transition metal dichalcogenides. *Nat. Phys.* 16, 1093–1096 2020).

I do consider the study as a comprehensive follow-up of the above-mentioned studies on a different moire system: 60-degree twisted TMD.

In the following, some suggestions the authors may consider incorporating in the revision:

1. Why is the 60-twisted TMD interesting? From my point of view, this system is special compared to the near zero-twist system because the lowest flat bands are folded from the gamma points. However, they do not host many exotic valley-related exotic phenomena due to the vanished spin-orbital coupling. I understand it is a simpler system for theoretical modeling, but it is not immediately obvious to me why this system is important.

2. Could the author further elaborate on the precision in the theoretical model? meV-scale precision is required to accurately identify the flat-band structure. Can the theoretical calculation achieve this? A following-up question: if there is spin-orbital coupling, can the calculation still catch the precise band structure?

3. Could the author further elaborate on the effect of graphene substrate, both on the structure reconstruction and the correlated phases? Why not fabricate the sample on a decoupling layer (eg. Li, H., Li, S., Naik, M.H. et al. Imaging moiré flat bands in three-dimensional reconstructed WSe₂/WS₂ superlattices. *Nat. Mater.* 2021)?

4. The manuscript mostly discusses the in-plane reconstruction, how about the out-of-plane deformation?

Reviewer #2 (Remarks to the Author):

This paper presents STM/STS measurements that visualize localized states in WSe₂ twisted bilayer samples over a range of twist angles. The results show a drastic change in behavior between bilayers far from 60° (unreconstructed) and bilayers within a few degrees of 60° (reconstructed). In particular, the samples near 60° show features indicative of an array of triangular quantum wells with several sharp energy levels with varying numbers of nodes, as expected for quantum wells. The authors also show excellent agreement between their results and a theoretical model which utilizes a transition from a moiré potential for twist angles far from 60° to a reconstruction induced confining potential for twist angles close to 60°.

I think this paper is very well written and presents a compelling combination of experiment and theory to explain the existence of flat bands in twisted WSe₂ structures that will be valuable to the field. The figures are well thought out and clear. In particular, I think the data and layout of Fig. 3 is outstanding.

I have a few relatively minor questions and comments listed below that I believe may make the paper even stronger than it already is. I strongly recommend this manuscript for publication after the authors consider these few minor points.

1. Throughout the manuscript, the authors refer to “larger angles” exhibiting reconstruction and sharp flat bands. Also, the authors often talk about phenomena occurring “when the twist angle is larger than 57°”. I wonder whether it might be more clear to change the language to instead refer to how close the angle is to 60°. For instance, “when the twist angle is within 3° of 60°.” I think something like this would be more precise. Also, I found it slightly confusing that the “larger angles” had reconstruction and sharp flat bands because many papers in the literature use “larger angles” to refer to samples with larger misalignment. I can see advantages to both ways of describing the angles, but I just wanted to bring this to the author’s attention in case they want to revise their language to be more clear.

2. In the first sentence on page 8, I think there is a typo where it says “Fig. 1b”. I think this should be “Fig. 1c”.

3. On page 8, when estimating onsite Coulomb repulsion, am should be defined. I did not see a definition in the manuscript.

4. For Fig. 4a, is it appropriate to draw the “no reconstruction” diagram with A and B domains? I would think that there would be continuous variation of stacking in the non-reconstructed case, so one cannot define discrete domains.

5. In Fig. 4c, why is the reconstructed confining potential composed of equilateral triangles rather than bowed triangles as shown in Fig. 4a and Fig. 4b?

6. In the “Theoretical Modeling...” section, is $V_1 = 50$ meV a fitting parameter? Or is there some other rationale for this choice?

7. One practical question about the STM methods: how were you able to “find” your samples in the STM without alignment marks?

Reviewer #3 (Remarks to the Author):

The manuscript “Lattice reconstruction induced multiple ultra-flat bands in twisted bilayer WSe_2 ” by Li et al. reports on the STM study of reconstruction effect in twisted bilayer WSe_2 with varied twist angle. The multiple ultraflat bands in twisted bilayer transition metal dichalcogenides have been theoretically predicted before. In this work, they first experimentally observed multiple ultraflat bands near valence band edge in homobilayer WSe_2 , and systematically investigated the physical origin of the flat bands. Considering the recent discoveries of correlated phases in twisted bilayer transition metal dichalcogenides, this work will be of fundamental interest to the research community in this field. Therefore, this manuscript can be eligible for publication in Nature Communications once the authors clarify the points raised below.

(1) Since the authors study the reconstruction effect to the flat bands, they need to theoretically compare the reconstructed case with the rigid one (without reconstruction in three dimension), and quantitatively investigate how the reconstruction affect the flat bands in such system with twist angle around 60° .

(2) In the continuum model, the authors use parameters correspond to the Gamma point of the Brillouin zone of AB stacking bilayer WSe₂ to explain the flat bands. They could double check with experiment to distinguish whether these flat band states come from Gamma or K points.

(3) The authors conclude “Fig. 1f is the calculated local density of states (LDOS) at four moire sites, which match very well with the measured site-specific dI/dV spectrums”. Yes, they can reproduce the spatial distributions of the LDOS quite well. However, in the Fig. 1f, there is a huge gap between the v1 and v2, which means the v1 is an isolated flat band. But in the Fig.1c, the v1 is not isolated. As the gap is relevant to investigate the correlated phases, the authors need to clarify this difference.

(4) As shown in Fig. 3, the calculated LDOS maps match quite well with the experimental ones. However, energy separation between flat bands are different. For example, the calculated energy separation in Fig. 3d is larger whereas in Fig. 3g and j are smaller. Why the energy separations are so different from the experimental ones?

(5) The authors write in page 16 “We notice there is a peak around -1.1 V at the center of A regions in 57° TB-WSe₂ (Fig.2b), besides the two B-confined states. This indicates the cooperative interactions between moire potential V_M and confining potential V_C ”. Here, I cannot follow the discussion. Why the authors link the states localized in the A region and higher energy peaks (-1.8 V and 1.3-2.0 V) with the moire potential? They need to provide more evidence. Moreover, the author may want to say “Fig. 2c” but not “Fig.2b” in the sentence.

Minor points:

(1) It is difficult to read the red letter “A” in Fig.1b.

(2) In the first line of page 8, where is the spectral peaks in Fig.1b.

(3) While the work here is insightful, the major results are related to the multiple flat bands detected in homobilayer WSe₂. However, given the fact that the reference 33 discovers multiple flat bands in heterobilayer WSe₂/WS₂, the authors may want to compare their results with the reference 33. For example, the tendency is the same or different between the homo and heterobilayers?

(4) There is a confusion about the meaning of the “lattice reconstruction”. For example, in the section “Electronic structure without lattice reconstruction in the sample with 54.1° twist angle”, the authors conclude that the 54.1° system has no reconstruction. However, in Fig.1c, the v_1 is localized in the B region. As discussed in a published paper (Phys. Rev. Lett.121,266401(2018)), the spatial localization of v_1 change from O region to B region is due to the lattice reconstruction.

Point-by-point response to the comments from the reviewers

The responses to the reviewers' comments are in highlighted in blue below. The changes to the text are highlighted in red.

Response to Reviewer 1#

Comment: The manuscript contains interesting information on a timely topic: the existence and characterization of flat bands in twisted transition metal dichalcogenide homo bilayers. The data quality is high, and the manuscript is clearly written. I am in favor of publishing it on Nature Communication.

With that said, I do not consider this study as a substantial breakthrough in the field, for the following reasons: 1) the sample is fabricated on a conductive graphene substrate, which is well known to quench the correlated phenomena. The quantum confinement like features in the gamma flat band has been previously reported in the CVD growth samples (eg, Pan, Y. et al. Quantum-confined electronic states arising from the moiré pattern of MoS₂–WSe₂ heterobilayers. Nano Lett. 18, 1849–1855 2018; Zhang, C. et al. Interlayer couplings, moiré patterns, and 2D electronic superlattices in MoS₂/WSe₂ hetero-bilayers. Sci. Adv. 3, e1601459 2017). @) Compare with these studies, the main highlight of the manuscript is the ability to control the twisted angle using mechanical transfer technique, which has also been reported previously (Zhang, Z. et al. Flat bands in twisted bilayer transition metal dichalcogenides. Nat. Phys. 16, 1093–1096 2020). I do consider the study as a comprehensive follow-up of the above-mentioned studies on a different moire system: 60-degree twisted TMD.

Response: We thank the reviewer for the positive remarks on the significance and quality of our work. We agree with the referee that conductive graphite substrate may provide screening to the electrons and weaken electron-electron interactions. However, the graphite substrate does not affect the lattice reconstruction and allowed us to unveil the lattice reconstruction induced multiple ultra-flat bands and the corresponding wavefunctions.

The referee is correct that compared to the previous studies on aligned heterobilayer MoS₂-WSe₂ and homobilayers with a fixed twist angle, our samples allow a comprehensive study on the twist angle dependence. We show the drastic change of moiré flat-band properties with twist angle. The key discovery of our work is experimental revelation of the effects of lattice reconstruction in moiré superlattice on the band structures of twisted bilayer WSe₂, which has not been addressed before. We hope that our study, which unveiled the energy, wavefunction and the physical origin of the ultraflat bands, provides a foundation for the further study of this interesting moiré system.

Comment: In the following, some suggestions the authors may consider incorporating in the revision:

1. Why is the 60-twisted TMD interesting? From my point of view, this system is special compared to the near zero-twist system because the lowest flat bands are folded from the gamma points. However, they do not host many exotic valley-related exotic phenomena due to the vanished spin-orbital coupling. I understand it is a simpler system for theoretical modeling, but it is not immediately obvious to me why this system is important.

Response: We thank the suggestion of the referee. The referee is right that the moiré bands near zero-twist WSe₂ have origins both from the Gamma point and the K points. Therefore, it is more difficult to interpret the experimental data and provide a clear understanding of the data using simple theoretical models. For the near 60-degree twist samples, both the energy of the bands and the wavefunctions can be understood from a simple theoretical model as pointed out by the referee.

It is worth to mention that although the reconstruction happens in either 0°- or 60°-twisted TMDs, calculations demonstrate that flat bands in 0°-twisted TMD mainly originate from inhomogeneous hybridization, regardless of the reconstruction [PRB 102, 075413 (2020)]. However, for the twist angle close to 60°, lattice reconstruction results in multiple dispersionless ultra-flat bands. Therefore, only 60°-twisted TMD exhibits the twist angle-dependent drastic change of flat-band properties, from un-reconstructed moiré structure to reconstructed structure, which has been observed by our STM/STS measurements.

In the near 60° twist cases, multiple energy-separated ultra-flat bands with bandwidths in the order of several meV were observed in this work. This is in sharp contrast to the 0-degree twist samples observed in the pioneer work [Nat. Phys. 16, 1093–1096 (2020)], in which flat bands have bandwidth about tens of meV and are mixed with other bands, as shown in Fig.R1 below.

Figure R1 (a) Fig.2b of [Nat. Phys. 16, 1093 (2020)] on the flat bands of near zero-degree TB-WSe₂. Note that both the Gamma point and the K valleys contribute to the flat bands. (b) Fig.2k of the main text. The multiple sharp dI/dV peaks correspond to multiple ultra-flat bands in 58.4-degree twist sample were observed.

As the bands have much narrower bandwidths and are well separated in energy, we expect that correlation effects in these ultra-flat bands are much stronger and interesting interaction induced correlation phases will arise. Therefore, we expect that the near 60-degree twist samples are very interesting.

2. *Could the author further elaborate on the precision in the theoretical model? meV-scale precision is required to accurately identify the flat-band structure. Can the theoretical calculation achieve this? A following-up question: if there is spin-orbital coupling, can the calculation still catch the precise band structure?*

Response: In our model, the shape and depth are based on the DFT calculations for homobilayer MoS₂ [PRB 102, 075413 (2020)] which is very similar system to homobilayer WSe₂. As shown below (Fig. 3a, d, g, j of the main text), the model describes the energy of the bands near the valence band edges extremely well for a wide range of angles (57 degree to 58.4 degree). Moreover, the calculated wavefunctions of the moiré bands also match the STS maps very well (Fig.1e-h, Fig.3 in the main text). Therefore, we are confident that our model can faithfully

describes the physics of homobilayer WSe_2 near the valence band maximum.

Concerning spin-orbit coupling (SOC), the energy splitting caused by SOC must be zero at the Gamma point due to time-reversal symmetry. Slightly away from the Gamma point, the SOC effect is expected to be small. As the moiré Brillouin zone is small (the zone edge to Gamma point distance is about $|k| \sim \frac{4\pi\theta}{3a} < 0.13 \text{ \AA}^{-1}$), the moiré bands near the valence band maximum are originated from bands with small SOC effects. Therefore, the effect of SOC is negligible in our study. But the referee is right that for bands far below the valence band maximum, SOC effects can be more important as they are originated from bands with larger k . However, the bands far below the valence band maximum are not the focus of our study.

Revised Fig. 3a, d, g, j in main text

3. Could the author further elaborate on the effect of graphene substrate, both on the structure reconstruction and the correlated phases? Why not fabricate the sample on a decoupling layer (eg. Li, H., Li, S., Naik, M.H. et al. Imaging moiré flat bands in three-dimensional reconstructed WSe_2/WS_2 superlattices. *Nat. Mater.* 2021)?

Response: Previous ARPES studies have directly shown the intrinsic band structure for either TMD monolayer or heterostructure on graphite (graphene) substrate [Sci. Adv. 3, e1601832 (2017), Nat. Nanotech. 9, 111–115 (2014), Phys. Rev. B 99, 045134 (2019)]. The observed hybridization between WSe_2 and graphene, are >3 eV below E_F , and the important WSe_2 bands near valence band edges are not affected [Sci. Adv. 3, e1601832 (2017)]. In addition, as shown in Fig. S1b, our STS measurements show a nearly intrinsic semiconducting bandgap (valence band maximum: -1.05 V, conduction band minimum: $+0.99$ V) with a rather weak doping effect from the substrate.

Secondly, we have not observed moiré pattern (around 1nm periodicity) caused by the mismatch of WSe₂ and graphite lattice, suggesting that the substrate plays a negligible role in the reconstruction of the bilayer WSe₂. For reference, in a previous work of TB-WSe₂ on graphene-BN substrate [Nat. Phys. 16, 1093–1096 (2020)], both graphene/hBN moiré pattern (11 nm) and the TB-WSe₂ moiré pattern are clearly visible in their STM images. We therefore conclude that in our samples, the substrate-WSe₂ interaction is weaker than the referenced work.

We thank the reviewer suggesting the method used in the reference (Nat. Mater. 2021). This method is of great advantages because it allows controlling the carrier density of twisted bilayers by gate voltage. Currently, we are working along this direction.

4. The manuscript mostly discusses the in-plane reconstruction, how about the out-of-plane deformation?

Response: In a bilayer TMD system, the out-of-plane deformation is known to result in inhomogeneous interlayer hybridization due to the variation in the interlayer spacing [Phys. Rev. Lett. 121, 266401(2018)]. In the 54.1°-twist case, the moiré potential $V_M(\mathbf{r})$ is caused by this interlayer hybridization. As the twist angle further approaches 60°, the in-plane structural reconstruction plays a dominant role in modulating the electrons in the bands while the out-of-plane deformation becomes relatively insignificant [Phys. Rev. B 102, 075413 (2020)]. In our theoretical model, we gradually reduce the proportion of $V_M(\mathbf{r})$ to match our experimental results. In revised manuscript, we have taken the reviewer's suggestions and added the related descriptions in method part:

“...moiré potential term. And \mathbf{g}_i are moiré reciprocal vectors. Such a potential is produced by the out-of-plane deformation induced inhomogeneous interlayer hybridization. We choose a typical...”

“...In a reconstructed moiré structure, the combined effect of reconstruction-induced $V_C(\mathbf{r})$ and inhomogeneous hybridization-induced $V_M(\mathbf{r})$ results in the flat-band formation...”

Response to Reviewer 2#

This paper presents STM/STS measurements that visualize localized states in WSe₂ twisted bilayer samples over a range of twist angles. The results show a drastic change in behavior between bilayers far from 60° (unreconstructed) and bilayers within a few degrees of 60° (reconstructed). In particular, the samples near 60° show features indicative of an array of triangular quantum wells with several sharp energy levels with varying numbers of nodes, as expected for quantum wells. The authors also show excellent agreement between their results and a theoretical model which utilizes a transition from a moiré potential for twist angles far from 60° to a reconstruction induced confining potential for twist angles close to 60°.

I think this paper is very well written and presents a compelling combination of experiment and theory to explain the existence of flat bands in twisted WSe₂ structures that will be valuable to the field. The figures are well thought out and clear. In particular, I think the data and layout of Fig. 3 is outstanding.

I have a few relatively minor questions and comments listed below that I believe may make the paper even stronger than it already is. I strongly recommend this manuscript for publication after the authors consider these few minor points.

Response: We thank the reviewer for his/her very positive evaluation of our work.

1. Throughout the manuscript, the authors refer to “larger angles” exhibiting reconstruction and sharp flat bands. Also, the authors often talk about phenomena occurring “when the twist angle is larger than 57°”. I wonder whether it might be more clear to change the language to instead refer to how close the angle is to 60°. For instance, “when the twist angle is within 3° of 60°.” I think something like this would be more precise. Also, I found it slightly confusing that the “larger angles” had reconstruction and sharp flat bands because many papers in the literature use “larger angles” to refer to samples with larger misalignment. I can see advantages to both ways of describing the angles, but I just wanted to bring this to the author’s attention in case they want to revise their language to be more clear.

Response: We have taken the reviewer's suggestions and have revised the related descriptions as follows:

Abstract part:

"...we show the emergence of multiple ultra-flat bands in twisted bilayer WSe₂ when the twist angle is within 3° of 60°."

Introduction part:

"These ultra-flat bands, with bandwidths estimated to be a few meV only, appear when the twist angle is within 3° of 60°."

"the ultra-flat bands mainly arise from atomic reconstruction of the moiré superlattices when the twist angle is within 3° of 60° due to the enhanced interlayer interaction."

Main part:

"Lattice reconstructions at twist angle within 3° of 60°:"

"twist angles far from 60° (for instance 54.1°),"

2. In the first sentence on page 8, I think there is a typo where it says "Fig. 1b". I think this should be "Fig. 1c".

Response: We thank the reviewer's careful checks and have fixed the typo:

"To understand the origin of the spectral peaks near the VB edge depicted in Fig. 1c,..."

3. On page 8, when estimating onsite Coulomb repulsion, am should be defined. I did not see a definition in the manuscript.

Response: We have taken the reviewer's suggestions and added more description as follows:

"For 54.1°-twist TB-WSe₂, we estimate the on-site Coulomb repulsive interaction U by using $U = e^2/(4\pi\epsilon d) \approx 180\text{meV}$. Here $d \approx 2\text{nm}$ is the spatial extent of the localized wave function, determined from dI/dV maps in Fig. 1e, and $\epsilon = 4$ is the in-plane dielectric constant. Such a reasonably large ratio for $U/\text{bandwidth}$ suggests the possibility of correlated insulator phase at half-filling or fractional-filling of the first moiré band^{25, 34}."

4. For Fig. 4a, is it appropriate to draw the "no reconstruction" diagram with A and B domains? I would think that there would be continuous variation of stacking in the non-reconstructed case, so

one cannot define discrete domains.

Response: Yes, it is appropriate to draw the “no reconstruction” diagram in Fig. 4a, based on previous DFT calculations [Phys. Rev. B. 98, 224102 (2018); Nat. Nanotechnol. 15, 592-597 (2020); Phys. Rev. B 102, 075413 (2020)]. We agree with the Reviewer that the A/B/O stacking in the non-reconstructed varies continuously. As the revised Fig. 4a-b shown in below, we have taken the reviewer’s suggestions and modified the diagram with dashed lines to mark the boundaries, to highlight the gradual variation of the A/B/O regions.

5. In Fig. 4c, why is the reconstructed confining potential composed of equilateral triangles rather than bowed triangles as shown in Fig. 4a and Fig. 4b?

Response: The bowed triangles shown in Fig. 4a are topographic configurations of the moiré superlattice (for example: interlayer separation landscape), which corroborate the STM topography shown in Fig. 4b. According to the calculations reported in [PRB 102, 075413 (2020)], the confining potential originates from in-plane relaxations, therefore, the potential shape is associated with the inhomogeneous strain distribution in the reconstructed moiré structure.

6. In the “Theoretical Modeling...” section, is $V_1 = 50$ meV a fitting parameter? Or is there some other rationale for this choice?

Response: The parameters in the reconstructed moiré potential is mainly based on previous DFT calculations in [PRB 102, 075413 (2020)]. In our calculations, slightly modifying the model parameters will change their energy separations but the spatial distribution behaviors remain. In revised manuscript, we have further optimized parameters in our theoretical model. As shown by the revised Fig. 3a, d, g, j below, calculated energy separations between flat-band states match very well with the experimental values.

Revised Fig. 3a, d, g, j in main text

7. One practical question about the STM methods: how were you able to “find” your samples in the STM without alignment marks?

Response: We have made some marks around the TB-WSe₂ sample region on the graphite substrate. These marks are visible in a camera mounted in the STM system. Guided by these marks, we could position the STM tip at the TB-WSe₂ samples. We thank the Referee for the question and we now added this information in the Methods.

Response to Reviewer 3#

The manuscript “Lattice reconstruction induced multiple ultra-flat bands in twisted bilayer WSe₂” by Li et al. reports on the STM study of reconstruction effect in twisted bilayer WSe₂ with varied twist angle. The multiple ultraflat bands in twisted bilayer transition metal dichalcogenides have been theoretically predicted before. In this work, they first experimentally observed multiple ultraflat bands near valence band edge in homobilayer WSe₂, and systematically investigated the physical origin of the flat bands. Considering the recent discoveries of correlated phases in twisted bilayer transition metal dichalcogenides, this work will be of fundamental interest to the research community in this field. Therefore, this manuscript can be eligible for publication in Nature Communications once the authors clarify the points raised below.

Response: We thank the reviewer very much for his/her appreciation of the significance and quality of our work and all the positive comments and very useful suggestions that have allowed us to improve our manuscript. Specific point-to-point responses are as follows:

(1) Since the authors study the reconstruction effect to the flat bands, they need to theoretically compare the reconstructed case with the rigid one (without reconstruction in three dimension), and quantitatively investigate how the reconstruction affect the flat bands in such system with twist angle around 60°.

Response: We thank the suggestion of the referee. We now added the calculations in the SI (Fig.S5) for the rigid rotation case with 58.4° twist angle without considering lattice reconstruction. As shown in Fig. R1 a,b, the flat band features, both spatial localization and wavefunction patterns, are inconsistent with the observed multiple B-confined states in Fig.R1c (Fig. 3I in main text).

Figure R1. Calculated band structure (a) and LDOS maps (b) of 58.4° TB-WSe₂ based on moiré

potential $V_M(\mathbf{r})$. Note that the valence band edge is set at $E=0$. (c) The experimentally observed LDOS map of wavefunctions for the first six states in 58.4° TB-WSe₂ (Fig.3I in main text).

In our theoretical model, moiré potential $V_M(\mathbf{r})$ originates from inhomogeneous interlayer hybridization, produced by the out-of-plane deformation. The triangular potential $V_C(\mathbf{r})$ is from the in-plane structural reconstruction. The combined effect of reconstruction-induced $V_C(\mathbf{r})$ and interlayer hybridization-induced $V_M(\mathbf{r})$ results in the flat-band formation in reconstructed moiré structure. Therefore, in our calculations for 57° - 58.4° cases, we have adjusted the parameters (V_0, V_1) in $((V_M(\mathbf{r}) + V_C(\mathbf{r}))$ to fit the observed flat-band structures. As shown in Fig. S6, the reconstruction-induced $V_C(\mathbf{r})$ increases and inhomogeneous hybridization-induced $V_M(\mathbf{r})$ decrease as the twist angle approach 60° . This tendency agrees with previous DFT calculations, showing the enhanced reconstruction-induced potential with the twist angle (Fig. R2) [Phys. Rev. B 102, 075413 (2020)].

Figure R2. DFT calculated confining potential for 57.35° (a) and 58° from [Fig.11 in Phys. Rev. B 102, 075413 (2020)].

(2) In the continuum model, the authors use parameters correspond to the Gamma point of the Brillouin zone of AB stacking bilayer WSe₂ to explain the flat bands. They could double check with experiment to distinguish whether these flat band states come from Gamma or K points.

Response: We thank the reviewer's suggestions. We have added the height-dependent dI/dV measurements in revised SI. Previous studies showed that the STS peaks originated from K-point states in TMDs, decay faster with decreased tip-sample distance (d) [Li, H., Li, S., Naik, M.H. et al. Imaging moiré flat bands in three-dimensional reconstructed WSe₂/WS₂ superlattices. Nat. Mater. (2021)]. As shown in Fig. S1d, the v1 peak in our data show almost same decay constant as the peak near $-1.8V$. Since the peak near $-1.8V$ is from Γ points of WSe₂ [Nat. Phys. 16, 1093-1096 (2020)], same decay behavior evidences v1 peak corresponds to electronic states at Γ

points. So, we can make a conclusion that the flat band states come from Gamma point in our experiment.

Figure S1d. Tip-sample distance (d)-dependent STS at the B site ($V=-2.0$ V, $I= 100$ pA, 300 pA, 500 pA, 800 pA, 1.0 nA, 1.2 nA, 1.5 nA, 1.7 nA and 2.0 nA) in 54.1° TB-WSe₂.

(3) The authors conclude “Fig. 1f is the calculated local density of states (LDOS) at four moire sites, which match very well with the measured site-specific dI/dV spectrums”. Yes, they can reproduce the spatial distributions of the LDOS quite well. However, in the Fig. 1f, there is a huge gap between the $v1$ and $v2$, which means the $v1$ is an isolated flat band. But in the Fig. 1c, the $v1$ is not isolated. As the gap is relevant to investigate the correlated phases, the authors need to clarify this difference.

Response: The absence of the energy gap in dI/dV spectra is due to the experimental broadening during STS measurement. The Gaussian fitted FWHM is ~ 37 mV for $v1$ peak in site B, and ~ 55 mV for $v2$ peak in site A. Based on the method described in Supplementary Section 2, the deconvoluted intrinsic bandwidths are ~ 15 mV for $v1$ and ~ 22 mV for $v2$. The measured dI/dV spectra and deconvoluted intrinsic $v1$ and $v2$ peaks are shown in Fig. S3 and also revised Fig. 1c, showing a nearly isolated $v1$ peak. Moreover, the calculated LDOS at B/A site show an energy separation of ~ 40 meV between $v1$ and $v2$, matching well with experimental result (~ 36.5 mV). We thank the suggestion of the referee and we added a paragraph in the Supplementary Information to clarify this point.

Figure S3. dI/dV spectrums at the B/A sites and deconvoluted intrinsic $v1/v2$ peaks.

(4) As shown in Fig. 3, the calculated LDOS maps match quite well with the experimental ones. However, energy separation between flat bands are different. For example, the calculated energy separation in Fig. 3d is larger whereas in Fig. 3g and j are smaller. Why the energy separations are so different from the experimental ones?

Response: We thank the referee for this valuable question which helped us to improve the quality of the paper. The discrepancy is mainly due to the simplified model used in our numerical calculation which already captures most of the features of the energy of the bands and of the wavefunctions. We used a triangular quantum well with vertical potential walls ($V_C(\mathbf{r})$) to simulate the states in 58.4° case. However, as the DFT calculated confining potential shown in Fig. R2 [PRB 102, 075413 (2020)], the triangular potentials are not formed by sharp vertical walls. The potential varies more smoothly. Therefore, we further optimized the reconstruction-induced $V_C(\mathbf{r})$ with more smooth boundaries, as shown in Fig.R3 below. In revised manuscript, we have recalculated the flat-band structures for all the reconstructed cases based on the updated $V_C(\mathbf{r})$. As the revised Fig. 3a, d, g, j shown below, calculated energy separations between flat-band states match very well with the experimental values.

Figure R3. (a) Reconstruction-induced $V_C(\mathbf{r})$ we used in previous calculation (left) and revised manuscript (right). The left plot shows a sharp transition between B and A domains. The right plot shows a smooth transition between B and A domains. (b) The corresponding potential profile along the line O-B-A-O

Revised Fig. 3a, d, g, j in main text.

(5) The authors write in page 16 “We notice there is a peak around -1.1 V at the center of A regions in 57° TB-WSe₂ (Fig.2b), besides the two B-confined states. This indicates the cooperative interactions between moire potential V_M and confining potential V_C ”. Here, I cannot follow the discussion. Why the authors link the states localized in the A region and higher energy peaks (-1.8 V and 1.3-2.0 V) with the moire potential? They need to provide more evidence. Moreover, the author may want to say “Fig. 2c” but not “Fig.2b” in the sentence.

Response: We agree with the reviewer that the meaning of this discussion is not clear. We also agree that the peak around -1.1 V cannot evidence the coexistence of moiré potential $V_M(\mathbf{r})$ and $V_C(\mathbf{r})$ directly. Since this discussion is not relevant to the key results of this work, we decided to

delete this sentence in the revised manuscript.

Minor points:

(1) It is difficult to read the red letter “A” in Fig.1b.

Response: We have taken the reviewer’s suggestions and changed all the letters to black.

(2) In the first line of page 8, where is the spectral peaks in Fig.1b.

Response: We thank the reviewer’s careful checks and it is a typo:

“To understand the origin of the spectral peaks near the VB edge depicted in Fig.1c,....”

(3) While the work here is insightful, the major results are related to the multiple flat bands detected in homobilayer WSe₂. However, given the fact that the reference 33 discovers multiple flat bands in heterobilayer WSe₂/WS₂, the authors may want to compare their results with the reference 33. For example, the tendency is the same or different between the homo and heterobilayers?

Response: We thank the referee for the comment. Heterobilayer WSe₂/WS₂ is indeed a very interesting system which is under intense study currently. Many correlated phases such as the Mott insulating phase, anti-ferromagnetic phases and Wigner crystals at fractional fillings were observed. However, in heterobilayer WSe₂/WS₂, only a single sharp peak, corresponding to the two moiré bands near the valence band maximum were observed (The two top moiré bands overlap in energy). While in homobilayer WSe₂ as studied in our work, multiple ultraflat bands which are well separated in energy were observed. We hope that our study, which unveiled the energy, wavefunction and the physical origin of the flat bands, provides a foundation for the further study of this interesting moiré system.

(4) There is a confusion about the meaning of the “lattice reconstruction”. For example, in the section “Electronic structure without lattice reconstruction in the sample with 54.1° twist angle”, the authors conclude that the 54.1° system has no reconstruction. However, in Fig.1c, the v1 is localized in the B region. As discussed in a published paper (Phys. Rev. Lett.121,266401(2018)), the spatial localization of v1 change from O region to B region is due to the lattice reconstruction.

Response: The lattice reconstruction in this manuscript is referred to in-plane structural reconstruction, which only happens with the twist angle very close to 0° or 60° [Phys. Rev. B. 98, 224102 (2018), Phys. Rev. Lett. 124, 206101 (2020), Nat. Nanotechnol. 15, 592-597 (2020)]. The 54.1° twist system undergoes out-of-plane deformation but no significant in-plane reconstruction. The out-of-plane deformation gives the moiré potential $V_M(\mathbf{r})$ in our model calculation. In the mentioned papers, the spatial transformation of flat-band localization upon relaxation is also due to the out-of-plane deformation. We have changed the related description to make it clear, according to the referee's comments:

Method-Theoretical Modeling part:

“... \mathbf{g}_i are moiré reciprocal vectors. Such a potential is produced by the out-of-plane deformation induced inhomogeneous interlayer hybridization. We choose...”

Main part:

“...For the 54.1° sample, its potential term $V_M(\mathbf{r})$ is from inhomogeneity of the interlayer hybridization due to out-of-plane deformation in the moiré superlattice...”

“...displayed in Fig.4c-d, without this in-plane structural reconstruction (54.1°), the...”

“...The gradual variation of the two potentials indicates a transition from the interlayer hybridization dominating to reconstruction-induced confining potential dominating....”

Fig. 4a caption:

“... a, Illustration of the moiré pattern with and without twist angle-dependent reconstruction (right and left, respectively)”

REVIEWERS' COMMENTS

Reviewer #1 (Remarks to the Author):

The authors have satisfactorily addressed all my previous concerns. I support the publication of the manuscript.

One more minor suggestion the author may consider: Authors present Fig S1d, the di/dv signal as a function of tip-sample separation to support that the flat bands are folded from the Gamma point. The authors should be careful about the tip-induced band bending in this kind of measurement. A brief discussion about how the authors rule out the tip effect will be very helpful.

Reviewer #2 (Remarks to the Author):

As I mentioned in my initial review, I think this paper presents a compelling combination of experiment and theory that will be of significant interest to the research community. The authors have adequately addressed the questions and comments from my initial review and I recommend this article for publication.

Reviewer #3 (Remarks to the Author):

The authors have mainly addressed my questions. I recommend the publication of this work in Nature Communications. By the way, I am very interested in one issue: In the measured dI/dV , for instance, the results in Fig.1c, did the authors already subtract the effect of the graphene substrate in these colored lines, or the substrate has no contribution to the measured dI/dV . If there has no contribution, probably because the graphene and TMD have a very large twist angle. Could the authors identify the angle between the graphene and TMD?

Point-by-point response to the comments from the reviewers

The responses to the reviewers' comments are highlighted in blue below. The changes to the text are highlighted in red.

Response to Reviewer 1#

Comment: The authors have satisfactorily addressed all my previous concerns. I support the publication of the manuscript.

One more minor suggestion the author may consider: Authors present Fig S1d, the di/dv signal as a function of tip-sample separation to support that the flat bands are folded from the Gamma point. The authors should be careful about the tip-induced band bending in this kind of measurement. A brief discussion about how the authors rule out the tip effect will be very helpful.

Response: We thank the reviewer for recommending publication of our manuscript on Nature communications.

We agree with the referee that tip-induced band bending (TIBB) can arise due to poor screening of electric fields at such semiconducting surface. In Fig S1d, the slight energy shift of v_1 and Γ_2 peaks can be attributed to TIBB-induced variation. Similar behaviors have been observed in TMD monolayer and TMD heterobilayer [Nat. Mater. 13, 1091–1095 (2014); Nat. Mater. 20, 945–950 (2021)]. As tip-sample separation decreasing, both v_1 and Γ_2 peaks in Fig S1d shift toward deeper energy, agreeing well with the tendency of Γ -point states reported TMD monolayer and heterobilayer.

For each sample, we have performed a series of STS measurements at several areas of twisted region. By comparing the STS results at different areas, we confirmed the consistency to ensure the accuracy of the findings.

In revised manuscript, we have taken the reviewer's suggestions and added the related descriptions as follows:

Supplementary information part:

“...Previous studies showed that the STS peaks originated from K-point states in TMDs decay faster with decreased tip-sample distance (d)^{3,4}. ... that the v_1 peak corresponds to electronic

states at Γ point. In addition, the slight energy shift of ν_1 and Γ_2 peaks can be attributed to tip-induced band bending (TIBB) effect, due to poor screening of electric fields at such semiconducting surface⁶. As tip-sample separation decreasing, both ν_1 and Γ_2 peaks in Supplementary Fig 1d shift toward deeper energy, agreeing well with the tendency of Γ -point states reported MoSe₂ monolayer⁴ and WSe₂/WS₂ heterobilayer³.

3. Li H, et al. Imaging moiré flat bands in three-dimensional reconstructed WSe₂/WS₂ superlattices. *Nat. Mater.* 20, 945–950 (2021).

4. Ugeda, M., Bradley, A., Shi, SF. et al. Giant bandgap renormalization and excitonic effects in a monolayer transition metal dichalcogenide semiconductor. *Nat. Mater.* 13, 1091–1095 (2014).

6. Feenstra, R. M. & Stroscio, J. A. Tunneling spectroscopy of the GaAs(110) surface. *J. Vac. Sci. Technol. B* 5, 923–929 (1987).

Method part:

“...The dI/dV spectra were collected by using the standard lock-in technique with a voltage modulation of 5-10 mV and frequency of 787.3 Hz. For each sample, STS measurements were performed at several areas of twisted region, to ensure the consistency and accuracy of the findings.”

Response to Reviewer 2#

As I mentioned in my initial review, I think this paper presents a compelling combination of experiment and theory that will be of significant interest to the research community. The authors have adequately addressed the questions and comments from my initial review and I recommend this article for publication.

Response: We thank the reviewer for recommending publication of our manuscript on Nature communications.

Response to Reviewer 3#

The authors have mainly addressed my questions. I recommend the publication of this work in Nature Communications. By the way, I am very interested in one issue: In the measured dI/dV, for instance, the results in Fig.1c, did the authors already subtract the effect of the graphene substrate in these colored lines, or the substrate has no contribution to the measured dI/dV. If there has no contribution, probably because the graphene and TMD have a very large twist angle. Could the authors identify the angle between the graphene and TMD?

Response: We thank the reviewer for recommending publication of our manuscript on Nature communications.

All the dI/dV spectra shown in manuscript, including STS maps, are the measured raw data without any substrate subtractions. Previous ARPES studies have directly shown the intrinsic band structure for either TMD monolayer or heterostructure on graphite (graphene) substrate [Sci. Adv. 3, e1601832 (2017), Nat. Nanotech. 9, 111–115 (2014), Phys. Rev. B 99, 045134 (2019)]. The observed hybridization between WSe₂ and graphene, are >3 eV below E_F, and the important WSe₂ bands near valence band edges are not affected [Sci. Adv. 3, e1601832 (2017)].

We have observed the moire structure formed by the graphite substrate and the monolayer WSe₂ region of sample. The moire periodicity will be: $\ell(\theta) = a_{HOPG} / \sqrt{1 + (\frac{a_{HOPG}}{a_{WSe_2}})^2 - 2(\frac{a_{HOPG}}{a_{WSe_2}})\cos\theta}$, this yields a maximum of $\ell \sim 0.97\text{nm}$ at $\theta = 0$. Fig.R1 shows the STM images at monolayer WSe₂ regions of two samples, showing a moire pattern of $\sim 0.8\text{ nm}$ and $\sim 0.94\text{ nm}$, respectively. The corresponding twist angles between the graphite and the monolayer WSe₂ are 11.5° and 4.1°, respectively. Since the HOPG is not single crystalline, it's difficult to control the relative orientation between WSe₂ and the underlying graphite when stacking.

As a widely used substrate for STM measurements, there is a number of research works about TMD layers on graphite substrate [Nat. Commun. 7, 10349 (2016); Nano Lett., 14, 2443–2447 (2014); 2D Mater. 2 034004 (2015)]. Such moire pattern with varied twist angle between TMD and graphite has been observed by STM, but STS measurements have not found any moire site-dependent features or significant difference at varied misorientation angle [Nat. Commun. 6,

6298 (2015).; Nano Research 11, 6102–6109 (2018)]. In conclusion, the graphite plays a negligible role in our measurements.

Figure R1. STM images of the moiré pattern between monolayer WSe₂ and graphite substrate. (a) moiré period: ~0.8 nm. (V=0.3 V, I=100 pA, 77K) (b) moiré period: ~0.94 nm. (V=-1.6 V, I=80 pA, 77K)